# Diagnosis and Management of Adult Malignancy-Associated Hemophagocytic Lymphohistiocytosis

**DOI:** 10.3390/cancers15061839

**Published:** 2023-03-18

**Authors:** Jerry C. Lee, Aaron C. Logan

**Affiliations:** Hematology, Blood and Marrow Transplantation, and Cellular Therapy Program, Division of Hematology/Oncology, University of California, San Francisco, CA 94143, USA; aaron.logan@ucsf.edu

**Keywords:** hemophagocytic lymphohistiocytosis, hemophagocytosis, HLH, mHLH, LAHS, ruxolitinib

## Abstract

**Simple Summary:**

Although rare, hemophagocytic lymphohistiocytosis (HLH), a syndrome of severe, dysregulated inflammation, is associated with poor survival when it occurs in conjunction with malignancy. This review discusses how published methods for diagnosing HLH can be applied in the setting of adult patients presenting with malignancy-associated HLH (mHLH) and offers evidence-based recommendations for the management of this clinically challenging scenario.

**Abstract:**

Hemophagocytic lymphohistiocytosis (HLH) is a syndrome of severe, dysregulated inflammation driven by the inability of T cells to clear an antigenic target. When associated with malignancy (mHLH), the HLH syndrome is typically associated with extremely poor survival. Here, we review the diagnosis of secondary HLH (sHLH) syndromes in adults, with emphasis on the appropriate workup and treatment of mHLH. At present, the management of HLH in adults, including most forms of mHLH, is based on the use of corticosteroids and etoposide following the HLH-94 regimen. In some cases, this therapeutic approach may be cohesively incorporated into malignancy-directed therapy, while in other cases, the decision about whether to treat HLH prior to initiating other therapies may be more complicated. Recent studies exploring the efficacy of other agents in HLH, in particular ruxolitinib, offer hope for better outcomes in the management of mHLH. Considerations for the management of lymphoma-associated mHLH, as well as other forms of mHLH and immunotherapy treatment-related HLH, are discussed.

## 1. Introduction

Hemophagocytic lymphohistiocytosis (HLH) is a life-threatening syndrome of excessive, dysregulated inflammation in response to a provoking trigger [1,2]. Defective termination of this immune response, driven by dysregulated positive feedback loops between CD8^+^ T lymphocytes and macrophages, results in hypercytokinemia that leads to cytolysis, tissue infiltration of immune cells, and end-organ injury. Frequently, HLH can cause death due to hemodynamic collapse and end-organ dysfunction.

The current understanding of HLH pathogenesis is derived from murine models and primary patient samples. In these studies, CD8^+^ T cells have been shown to be activated in response to an immunologic trigger, leading to the production of type 2 interferon (IFN), which primes macrophages to secrete additional proinflammatory cytokines (Figure 1) [3,4,5,6,7]. Deficiencies in this cytolytic pathway result in an inability to proceed with normal activation-induced cell death, generating uncontrolled accumulation and activation of CD8^+^ T cells, natural killer (NK) cells, macrophages, and proinflammatory cytokines [8]. When HLH occurs as the result of congenital deficiencies of key cytolytic pathway proteins, this is called primary HLH (pHLH), which mainly occurs in children [9,10]. In adults, HLH is usually driven by a highly immunogenic trigger (i.e., secondary HLH, sHLH) rather than primary cytotoxicity defects [11], though in some cases, a relevant, often hypomorphic genetic mutation affecting cell-mediated immunity may be identified [12,13,14,15].

Triggers for sHLH vary greatly by geographic location. Common triggers are diseases associated with immune activation, such as autoimmune disorders, infections, and malignancies (Table 1). In North America and Europe, around 50% of adult HLH is due to an underlying malignancy, with the remaining 50% associated with rheumatologic diseases (also known as macrophage activation syndrome, or MAS, in this setting), infections (especially chronic viral infections, such as Epstein–Barr virus (EBV), cytomegalovirus (CMV), varicella zoster virus (VZV), herpes simplex virus (HSV), or human immunodeficiency virus (HIV), and under less frequent circumstances, infections with other non-viral pathogens), or treatment, usually from cell therapy (hematopoietic cell transplantation (HCT) or chimeric antigen receptor (CAR) T cells) [17,18,19,20]. A minority of adult HLH cases are late presentations of primary HLH or are idiopathic, in which no discernable cause is found.

Based on retrospective data, outcomes in adults with sHLH differ widely between those in whom the syndrome is non-malignant (nmHLH) or malignancy-associated (mHLH). Long-term survival for adult nmHLH is comparable to pediatric HLH, whereas mHLH outcomes are particularly poor, with <20% survival at one year (median survival ~2 months) based on retrospective studies from the Mayo Clinic, MD Anderson Cancer Center, and the Harvard-affiliated hospitals [21,22,23]. mHLH therefore represents an area of much-needed investigation since many studies (for adult and pediatric HLH) exclude mHLH. Current management practices advise treating acute hypercytokinemia, followed by cancer-directed therapy [24,25,26,27]. It is unknown, however, whether treatment of mHLH confers a survival benefit over treatment of the underlying malignancy. It is also unknown whether the diagnostic criteria for mHLH should be the same as with nmHLH. This review provides an updated summary of the existing literature on the diagnosis and management of adult malignancy-associated HLH, especially in the setting of emerging research on effective treatment strategies in the age of engineered cellular and immunotherapies.

## 2. Diagnosis of Malignancy-Associated HLH

### 2.1. HLH-2004 Diagnostic Guidelines

Since HLH represents a spectrum of hyperinflammatory disorders with heterogeneous inciting conditions, often with both genetic and environmental components, diagnosis can be challenging. Traditionally, diagnosis is based on the HLH-2004 revised diagnostic guidelines from the Histiocyte Society, which do not distinguish between nmHLH and mHLH and rely on a combination of clinical, laboratory, and pathological data [28]. In patients without an HLH-predisposing genetic variant, five of eight diagnostic criteria are required (Table 2), some of which denote macrophage activation such as ferritin elevation and hemophagocytosis, and some of which denote T cell proliferation such as soluble IL2 receptor (sIL2R) [29,30].

These diagnostic guidelines were based on data from pediatric HLH, and differences between pediatric and adult HLH raise questions about their utility in adults. For instance, malignancy accounts for nearly 50% of adult triggers, as opposed to 8% in children [32,33]. There is a higher prevalence of hepatomegaly (95%) and neurological symptoms (33%) in pediatric cases compared with adults (18–67% and 9–25%, respectively) [34]. A ferritin of >2000 achieves a sensitivity of 70% and specificity of 68% in children [35], which rises to 90% sensitivity with 96% specificity if the ferritin is >10,000 ng/mL [36]. In adults, on the other hand, higher ferritin cutoffs have poorer test characteristics (at 3000 ng/mL, ~67% sensitivity/specificity [37]; at 10,000 ng/mL, 43% sensitivity [21]; at 50,000 ng/mL, <20% sensitivity/specificity [38]). This is due to a more heterogeneous range of conditions associated with hyperferritinemia in older individuals.

### 2.2. Alternatives to HLH-2004 Diagnostic Guidelines

Several attempts have been made to improve the HLH-2004 diagnostic guidelines. In addition to modifying ferritin cutoffs (or mandating hyperferritinemia as a required criterion due to its high negative predictive value), other suggestions have been made to redefine HLH around clinicopathologic features. In this view, the diagnosis of HLH is made based on parameters fulfilling categories representing predisposing immunodeficiency, significant immune activation, and/or abnormal immunopathology—an approach that may better reflect the current understanding of HLH pathophysiology [31]. A modified HLH-2004 strategy has been proposed to assist with earlier diagnosis, which requires three of four clinical findings (fever, splenomegaly, cytopenias, and hepatitis), plus at least one of four immunologic test abnormalities (hyperferritinemia, elevated serum soluble IL-2Rα (sIL2R), absent/markedly decreased natural killer (NK) cell function, or the presence of hemophagocytosis) [39]. In a 2012 survey of HLH experts, an iterative questionnaire was utilized to determine the clinical features thought most important in adult HLH, which were the presence of a predisposing disease, fever, organomegaly, cytopenias, elevated ferritin, elevated LDH, and hemophagocytosis [40]. A scoring system (“HScore”) was developed to define and predict the likelihood of adult HLH based on HLH-2004 weighted parameters, for which the optimal cutoff HScore value was 169 (sensitivity 93%, specificity 86%) [41]; notably, the study cohort included 137 cancer patients out of a total of 312 adults. A subsequent study compared the accuracy of HScore against HLH-2004 and found that although the HScore achieved a sensitivity and specificity of 90% and 79% for adults at initial presentation, performance dropped to 73% sensitivity and specificity (similar to HLH-2004) when clinical status worsened [42].

The additional diagnostic challenges posed by mHLH are the many alternative explanations for abnormal laboratory parameters. Most mHLH is driven by hematologic malignancies, particularly lymphoma [43], where marrow and/or spleen neoplastic infiltration can explain cytopenias and splenomegaly; fever often occurs from concomitant infections in immunocompromised hosts or from tumor fever; hypofibrinogenemia can be secondary to malignancy-related disseminated intravascular coagulation (DIC); and hyperferritinemia can result from tumor-related inflammation or transfusional iron overload. As such, there has been no accepted definition for mHLH, and consensus recommendations have advocated for using HLH-2004 as a diagnostic tool in conjunction with physician judgement as to whether the clinical phenotype observed is out of proportion to the malignancy alone [25,27]. A high sIL2R/ferritin ratio has been proposed as a marker for lymphoma-associated HLH (LAHS). In a retrospective analysis of 21 patients comparing nmHLH and LAHS, the mean sIL2R/ferritin ratio was 0.66 amongst nmHLH patients and 8.56 in LAHS, hypothesized to reflect excessive T cell activation out of proportion to macrophage stimulation [44]. LAHS also have comparatively higher mean levels of microRNA-133 [45], IFN-inducible protein 10 (IP-10)/CXCL10, and monokine-induced by IFNγ (MIG)/CXCL9 [46], which have been proposed but not yet validated as diagnostic biomarkers.

In another retrospective analysis performed using the MD Anderson Cancer Center database, only 21% of patients with suspected HLH met HLH-2004 diagnostic criteria [22]. As such, the authors suggested expanding the diagnostic criteria to 18 variables, for which any five would be diagnostic. In this schema, 35 of 61 patients with pathologic hemophagocytosis or lymphohistiocytosis were thought to have true mHLH, with no differences in outcomes compared to those who met the standard HLH-2004 criteria, but with inferior outcomes compared to the remaining 26 patients with pathologic hemophagocytosis or lymphohistiocytosis but who did not meet the extended criteria.

Finally, a recent multicenter retrospective study using a cohort of 225 patients with hematologic malignancies for which sIL2R was available identified an optimal cutoff of sIL2R > 3601 U/mL and ferritin > 920 ng/mL, achieving a sensitivity of 88% and specificity of 76% for identifying the presence of HLH [47]. Comparatively, the cutoffs used in HLH-2004 (sIL2R > 2400 U/mL and ferritin > 500 ng/mL) demonstrate a sensitivity of 92% and a specificity of 72%. The authors suggest that the optimized HLH inflammatory (OHI) index, using sIL2R > 3900 U/mL and ferritin > 1000 ng/mL for simplicity, provides both diagnostic and prognostic value in hematologic malignancies with HLH.

These diagnostic modifications were proposed to enhance the expediency, simplicity, and generalizability to community practices. However, whether they are intended to raise suspicion index, increase accuracy, or reflect biological pathogenesis, any formal definition of adult mHLH will ultimately require prospective validation and international consensus.

## 3. Initial Workup of Adult HLH

Key to the diagnosis and management of HLH in adults is maintaining a high index of suspicion among patients with an unexpected hyperinflammatory clinical presentation—particularly patients with fever/sepsis, hyperferritinemia, and bi/pancytopenia [34]. Given the nontrivial turnaround time for several laboratory tests specified in HLH-2004, early suspicion and diagnosis are critical to allow for therapeutic intervention prior to the organ failure and death that commonly occur rapidly in adults with mHLH. Based on a review of the HLH management literature combined with our experience, we recommend a stepwise evaluation, starting with commonly available assessments that provide results quickly, followed by specialty testing (Figure 2).

As expected, there is an increased likelihood of an underlying malignancy with advanced age. A single center in Sweden estimated 1% of all adult hematologic malignancy patients developed HLH [48]. In Japan, a large survey of nearly 800 HLH patients demonstrated an underlying lymphoma in 68% of patients over age 60, compared to 38% between the ages of 30–59 and 10% between ages 15–29 [18]. Therefore, the possibility of underlying malignancy should be thoroughly evaluated and even expected, particularly for older adults presenting with HLH. Because the window between safely pursuing a tissue biopsy and clinical deterioration can be very short, earlier imaging and biopsy may be warranted.

## 4. Treatment of mHLH

### 4.1. HLH-94 Protocol

Since HLH is a common immunological pathway induced by a diverse spectrum of proinflammatory diseases, adequate control of HLH depends on the reversal of the underlying disease. The identification of this underlying disease is paramount to appropriate diagnostic evaluation and treatment. The challenge in mHLH, however, is several-fold. Often, the specific neoplasm causing HLH cannot be determined during acute cytokine storm, and delays in tissue diagnosis result in delays in adequate cancer-directed treatment. Additionally, HLH in this setting is often triggered by a combination of malignancy, infection, and/or predisposing immune dysregulation, requiring multifactorial and stepwise approaches to management. Standard treatment for HLH may also limit options for concurrent chemotherapy if high-dose steroids and etoposide cause severe myelosuppression, hepatic injury, infections, or impaired performance status. Finally, treatment of malignancy itself can cause HLH, as in the case of immunotherapy or cellular therapy.

There have not yet been any large prospective or randomized clinical trials published for adults with HLH. Current treatment strategies for adult HLH are based on pediatric trials. Primary HLH without treatment is uniformly fatal [50], and prior to the advent of etoposide-based therapy, outcomes were dismal, with 5-year overall survival (OS) of ~20% [51]. An effort led by the Histiocyte Society substantially improved mortality with the HLH-94 protocol by focusing on decreasing inflammation with a corticosteroid- and etoposide-based regimen, as well as reversing underlying triggers [31,52,53,54,55]. Over the past 30 years, improvements in this regimen have been attempted but have been incremental or inconclusive. The HLH-2004 protocol, which integrated cyclosporine into HLH-94, did not improve 5-year survival, which remained ~50–60% [28,54,56,57]. Allogeneic hematopoietic cell transplantation has been used to potentially cure primary or relapsed secondary HLH, but 20–30% of patients die before transplant, primarily in the first 8 weeks of therapy, from disease progression, infection, or bleeding—complications which, in some cases, may represent the consequences of treatment.

Given the relative effectiveness of HLH treatment in children, the conventional approach to treating sHLH (including, at least initially, mHLH) has been to suppress immune overactivation with HLH-94 therapy, with concurrent or subsequent treatment of the immune trigger, including cancer-directed therapy. HLH-94 consists of 8 weeks of dexamethasone, starting at 10 mg/m^2^/day, tapered by 50% every two weeks, with etoposide at 150 mg/m^2^ twice weekly for 2 weeks, then weekly for 6 weeks (Figure 3) [53]. If there is CNS involvement, intrathecal methotrexate at 12 mg/dose and intrathecal hydrocortisone at 15 mg/dose are given weekly until at least 1 week after resolution of CNS involvement, based on clinical and CSF indices.

### 4.2. Strategies to Treat mHLH

There is an ongoing debate over the role of HLH-94 in mHLH, primarily driven by the lack of prospective trials to guide management and in part due to the apprehension that dexamethasone and etoposide may interfere with further treatment of the underlying malignancy. Published treatment strategies suggest treating the underlying malignancy as soon as possible, with consideration of early steroid initiation to abrogate HLH-induced hypercytokinemia, followed by cancer-directed therapy (typically, cytotoxic chemotherapy) [25,26]. Some practitioners recommend against etoposide in this setting due to concerns for additive toxicities (particularly myelosuppression) and instead use steroids and well-tolerated adjunctive treatments only; some incorporate etoposide in chemotherapy regimens where possible (e.g., R-EPOCH for aggressive B cell lymphomas [58], SMILE for NK/T cell lymphomas [59]); others pursue a two-step approach with an HLH-94-based regimen, followed by chemotherapy at the time of organ recovery, which could be several weeks later. These decisions are made on a case-by-case basis, weighing whether the patient requires immediate inflammation-directed therapy or is sufficiently stable to receive chemotherapy directly. Importantly, adjunctive therapies during induction should also be considered, such as rituximab for concomitant active EBV infection [60] or intravenous immunoglobulin to treat hypogammaglobulinemia. Antimicrobial prophylaxis (for VZV/HSV, fungal, *Pneumocystis*) should be strongly considered.

If additional anti-cytokine treatment is required, but etoposide is not preferred, anti-IL1 and -IL6 agents have been utilized. The recombinant IL1 receptor antagonist (IL1Ra), anakinra, is approved by the US Food and Drug Administration (FDA) in autoimmune conditions such as rheumatoid arthritis and cryopyrin-associated periodic syndromes and by the European Medicines Agency (EMA) for systemic juvenile arthritis and adult-onset Still’s disease. Given the experience in rheumatologic disorders, anakinra has been used effectively in MAS [61,62], though its efficacy in MAS is markedly higher than in mHLH [63,64,65]. Naymagon, for instance, noted in a retrospective study of outcomes associated with anakinra treatment that those with rheumatologic condition-associated MAS experienced 75% OS, versus only 17% survival in patients with other underlying causes of sHLH [63]. Publications describing the use of anakinra for mHLH have been varied, ranging from frontline monotherapy use in Hodgkin lymphoma to relapsed/refractory HLH in myelodysplastic syndrome (MDS) and T cell lymphoma (TCL) [66,67,68]. The use of anti-IL6 therapy for adult HLH (with tocilizumab; no published data using siltuximab) has been extrapolated from its use in cytokine release syndrome (CRS) and COVID-19. Only a few publications have reported the efficacy of tocilizumab treatment in mHLH, with a recent retrospective review suggesting that the use of tocilizumab could increase infectious complications compared to conventional therapy [69,70].

### 4.3. Role of Allogeneic Transplantation

In adults with sHLH, idiopathic HLH, or mHLH responding to effective cancer therapy, patients who can be weaned off dexamethasone and etoposide without recurrence, have recovered normal immune function, and have reversed or controlled their underlying HLH trigger, can typically be monitored with serial assessments for markers of HLH activity, including ferritin, sIL2R, and chemistries [31]. Patients who do not meet these criteria, who developed CNS HLH, or who have predisposing gene mutations should undergo human leukocyte antigen (HLA) typing for consideration of allogeneic HCT. This includes those with mHLH with independent indications for allogeneic HCT based on the underlying malignancy, as well as those with mHLH not resolving with cancer therapy. If HLH continues to be active, dexamethasone/etoposide and/or adjunctive therapies may be continued as bridging therapy to the time of transplant. Myeloablative (MAC) and reduced-intensity conditioning (RIC) regimens have been directly compared in pediatric HLH; 14 patients at Cincinnati Children’s achieved a 3-year survival of 43% with MAC, and 26 patients achieved a 92% survival with RIC [71]. More recent studies in adult HLH have shown modest successes, achieving an overall survival of ~50% using the RIC regimen fludarabine and melphalan 100 mg/m^2^ [72] and 75% using alemtuzumab “pre-conditioning” prior to RIC [73]. Based on the limited available data, we favor the use of RIC approaches in adult patients with sHLH/mHLH proceeding to allogeneic HCT, and we include alemtuzumab if HLH is active proximal to the initiation of transplant conditioning following the approach of Gooptu et al. [73].

### 4.4. Biologic Therapy

For relapsed/refractory mHLH not responding to cancer therapy, consideration can be given to the use of agents shown effective in salvage management of pHLH/sHLH, and selection may be directed by what may be most compatible or synergistic with the selected cancer-directed therapy [74]. The anti-CD52 antibody alemtuzumab yielded a 64% response rate in the salvage setting, with 77% of pediatric patients proceeding to HCT, though notably with a high incidence of infections, a known complication from anti-CD52 therapy [75]. In China, adult refractory HLH has been treated with alemtuzumab along with the DEP regimen, consisting of liposomal doxorubicin, etoposide, and methylprednisolone, achieving complete responses in 27% and partial responses in 49% of patients [76]. Encouraged by these findings, a phase 2 clinical trial (NCT02385110) of adults with newly diagnosed or relapsed/refractory HLH (including mHLH) is currently underway, which uses alemtuzumab or tocilizumab combined with etoposide and dexamethasone.

Given the critical role of IFNγ in the pathogenesis of HLH, a prospective trial evaluated the IFNγ inhibitor emapalumab in 27 relapsed/refractory and 7 newly diagnosed pediatric patients with primary HLH and found an overall response rate of 65%, with 70% proceeding to HCT [77]. This led to FDA approval as salvage therapy for primary HLH without age restrictions, though notably, no adults were enrolled in this study, and the median age was 1 year old, limiting the generalizability of results to adult patients. A phase 2/3 study using emapalumab in adults (NCT03985423) was stopped due to sponsor withdrawal, and all 7 enrolled patients did not complete the study. Recently, a case series from Memorial Sloan Kettering was presented using emapalumab in 10 patients with heavily pretreated relapsed/refractory lymphoma and HLH; only 5 patients survived for more than 3 days [78]. All patients died within 1 month, suggesting IFNγ blockade is not an effective strategy in adults with refractory HLH. We do not currently employ emapalumab in adults with sHLH (including mHLH), though we have found it to be beneficial in adults with late-onset pHLH.

### 4.5. Ruxolitinib

Arguably the most promising agent currently under investigation in HLH is the Janus kinase (JAK) inhibitor ruxolitinib, which is a potent and selective inhibitor of JAK1 and JAK2 and a more modest antagonist of TYK2 (tyrosine kinase 2) and JAK3. Because many cytokines elevated in HLH signal via the JAK-STAT pathway [3,4,79,80,81], there are several preclinical and early clinical studies that indicate JAK-STAT inhibition may decrease immune hyperactivation and improve patient outcomes, including evidence that ruxolitinib may restore the sensitivity of CD8^+^ T cells to steroid-induced apoptosis [82]. Many case reports and pilot studies have now been published regarding the treatment of newly diagnosed or relapsed/refractory adult HLH [83], including those with mHLH, using ruxolitinib (Table 3). In these published reports with available data, clinical manifestations tend to resolve rapidly, with many patients showing improvements in platelet count, ferritin, soluble IL-2 receptor, AST, ALT, and fibrinogen levels within 7–14 days. These data are challenging to interpret, however, in the setting of methodological and disease heterogeneity, differing timepoints for which ORR and OS are assessed, and publication bias.

In several studies, the incorporation of ruxolitinib in the frontline setting was paired with dose reductions of etoposide and/or dexamethasone. Stalder et al. tested the use of ruxolitinib as an HLH-94 sparing strategy, allowing patients to receive between 50 and 150 mg/m^2^ etoposide initially and 10 mg/day of dexamethasone [86]. Wang et al. used etoposide at 100 mg/m^2^ once weekly with liposomal doxorubicin [91]. Indeed, this etoposide-sparing approach has been studied in the pediatric setting, in which newly diagnosed patients receive ruxolitinib with or without methylprednisolone and only receive an HLH-94-based regimen if there was an unfavorable response within a four-week induction period [92]. A similar trial in the United States for frontline and relapsed/refractory pediatric HLH is currently enrolling (NCT04551131). These approaches suggest that cytokine-directed therapy using minimally myelosuppressive medications such as ruxolitinib could be used in mHLH to temper the HLH cytokine storm, thus relieving the need for intensive etoposide-based regimens and bridge patients to cancer-directed therapy.

### 4.6. Investigational Agents

Other early-phase clinical trials for adult mHLH include investigating existing agents with known efficacy in hematologic malignancies with broad activity, such as venetoclax (NCT05546060) or zanubrutinib (NCT05320575). For EBV-associated HLH such as post-transplant lymphoproliferative diseases (PTLD) and EBV+ sarcomas, a phase 2 trial of the investigational agent tabelecleucel, an allogeneic, off-the-shelf EBV-specific T cell immunotherapy, is currently active (NCT04554914) based on efficacy demonstrated in immunochemotherapy-refractory EBV+ PTLD [93]. Another investigational agent under evaluation is ELA026, a human monoclonal immunoglobulin G1 signal regulatory protein (SIRP)-directed antibody (NCT05416307). SIRPα has long been implicated in the pathogenesis of HLH due to hematopoietic stem cell (HSC) surface downregulation of CD47, which normally interact with SIRPα to prevent autophagy [94]. In this case, ELA026 is thought to rapidly induce the phagocytosis of myeloid-derived antigen-presenting cells and pathogenic CD8^+^ T cells in sHLH and is currently being tested in a phase 1b dose-escalation study using monocyte depletion as a biomarker. Finally, in a preclinical study, inhibition of type II protein arginine methyltransferase (PRMT5) has shown significant potential in murine models of sHLH, consistent with its known role as an important mediator of inflammatory T cell subsets in similar disease contexts of autoimmunity and graft-versus-host disease (GVHD) [88].

## 5. Special Populations and Treatment-Related HLH

### 5.1. Lymphoma-Associated HLH (LAHS)

Lymphoma is the most well-characterized and common underlying cause of mHLH, as HLH can occur in up to 20% of patients with certain lymphoma histologies (e.g., intravascular B cell or nasal NK/T lymphomas) [48]. In fact, lymphoma may account for a third of all adult HLH [17], which increases up to two-thirds in patients older than 65 years [18,95]. HLH in lymphoproliferative disorders is often multifactorial, given their association with chronic immunomodulating viruses such as EBV, HIV, or HHV-8 (also known as KSHV or Kaposi sarcoma [KS]-associated herpesvirus), which are independently proinflammatory and cause other cytokine storm syndromes such as chronic active EBV, Castleman disease, and KSHV inflammatory cytokine syndrome (KICS). In lymphoma, clonally transformed neoplastic cells can exhibit direct cytokine secretion (particularly T cells) [96], resulting in HLH, while B cell neoplasms also tend to increase IFN-γ secretion either by association with mature T cells or responses to viral infections/reactivations. While HLH is more likely to occur in poor prognosis lymphomas, particularly high-grade or intravascular B cell and NK/T cell lymphomas, it is unknown whether HLH in this setting is purely indicative of cancer severity or whether it independently worsens mortality. If the latter is true, this implies that initial HLH-directed therapy may be more effective than cancer-directed therapy alone.

One of the first publications on LAHS reviewed nine peripheral T cell lymphomas (PTCL)-associated with hemophagocytic syndrome and compared them to known cases diagnosed by Scott and Robb-Smith, who first described “hemophagocytic medullary reticulosis” in 1939 [97]. In their clinicopathological review, the authors determined that T cell lymphoma-associated HLH shared many similarities to those described earlier by Scott and Robb-Smith—many of which have been retrospectively diagnosed with lymphoma [98]. Since the recognition that HLH and lymphoma do not infrequently occur together, several retrospective studies of LAHS have been published. In a cohort of 159 PTCL patients with and without HLH, 23% had LAHS, which was associated with a median survival of 3 months compared to 16 months in those without HLH [99]. Comparing LAHS to other adult HLH syndromes, a study from China demonstrated concordance with most features of the HLH-2004 diagnostic criteria, but the percentage of patients with severe hypofibrinogenemia, thrombocytopenia, and elevations in LDH were higher in LAHS [100]. Differences between T/NK cell LAHS and B cell LAHS have also been reported, with T/NK cell LAHS being younger and more likely to have DIC, bone marrow involvement, end-organ dysfunction, and worse survival [101,102]. Indeed, T/NK cell lymphomas associated with HLH have among the worst prognosis, with survival estimates approximately 1 month from presentation [103]. In this cohort of patients, those treated with pegaspargase were found retrospectively to have a longer median survival >100 days [104]; however, this likely reflects selection bias, as patients able to receive pegaspargase are likely less ill at presentation.

Certain B cell lymphomas are more prone to develop HLH. In fact, B cell HLH was originally proposed as an Asian variant of intravascular lymphoma, given its higher prevalence in East Asian countries [105,106,107,108], but B-LAHS is neither restricted to certain types of lymphoma nor to Asia. In a retrospective review of LAHS cases from France, roughly half of the cases were non-Hodgkin lymphoma (NHL), a quarter were Hodgkin lymphoma, and the remaining quarter were T cell lymphomas [109], which reflects differences in lymphoma epidemiology between the West and Asia. Among the 71 cases reported, 9% were not able to receive lymphoma treatment prior to death, and 27% died within 30 days. Nearly 20% did not receive treatment for HLH. Of those who were treated, ~50% achieved a CR after the first-line regimen. The median OS was ~6 months, with the best outcomes for HHV-8^+^ NHL; receipt of etoposide was associated with improved overall survival [109]. Indeed, several publications have suggested that LAHS should be treated with etoposide-containing chemotherapy regimens, but there are no large prospective studies confirming this approach [110,111]; other studies have demonstrated improved outcomes with autologous transplant or ruxolitinib [90,112]. Since etoposide has activity in lymphoma and in HLH, we advise incorporating etoposide when possible (such as dose-adjusted R-EPOCH or R-CHOEP regimens [113,114,115]), acknowledging there is no clear evidence supporting this practice.

### 5.2. Leukemia, Myelodysplastic Syndromes, and Allogeneic Transplantation

HLH occurring in association with acute and chronic leukemias, myelodysplastic syndrome (MDS), and myeloproliferative neoplasia is a rare occurrence, which has proven largely prohibitive for prospective studies in these patient populations. The identification of HLH in the setting of these disorders may be particularly complicated given the high prevalence of multilineage cytopenias, consumptive coagulopathy, and elevation of acute phase reactants at diagnosis, thus limiting the specificity of HLH-2004 diagnostic criteria in this setting. Most of the published literature available to provide treatment guidance derives from case reports and small case series. Stalder and colleagues recently reported a small single-center study in which patients with acute myelogenous leukemia (AML)-associated HLH received dose-adjusted ruxolitinib, etoposide, and dexamethasone (adRED) combined with induction chemotherapy [86]. Six patients (two with HLH at diagnosis of AML, four with HLH at relapse) who met HLH-2004 criteria for HLH diagnosis and had an HScore >169 were treated with adRED and exhibited a 100% overall response rate within one week. In the absence of other evidence, we endorse this approach in patients presenting with acute leukemia- or MDS-associated HLH.

Secondary HLH has also been reported in a small number of patients following allogeneic HCT [19,116,117,118], with an estimated incidence of 1% post-allogeneic HCT (0.15% for autologous HCT) [119]. Gower et al. reported findings consistent with HLH in 6.8% of recipients of 9/10 HLA-mismatched unrelated donors, with all HLH cases in recipients of low CD34^+^ count bone marrow grafts and none with peripheral blood stem cell grafts [120]. HLH was also previously reported to occur with relatively high incidence in adults receiving umbilical cord blood grafts, again with a finding that low CD34^+^ stem cell dose was a significant risk factor [121]. While the use of umbilical cord blood grafts is currently low in Europe and the United States (<5% of allografts) and is expected to decline further [122,123], the use of increasingly mismatched unrelated donors is anticipated with the use of post-transplant cyclophosphamide graft-versus-host disease (GVHD) prophylaxis [124]. The differential incidence of HLH with different GVHD prophylaxis approaches is unknown. Optimal treatment for HLH occurring after allogeneic HCT, regardless of the donor, remains uncertain, as there are essentially no published data to provide guidance. The diagnosis of sHLH after HCT can also be challenging to distinguish from allogeneic engraftment syndrome, CRS after haploidentical transplantation (“haplostorm”), acute GVHD, and severe sepsis [125,126,127]. Given the potential overlap with GVHD phenomena [128], the diagnosis may be challenging, and we currently favor treatment incorporating corticosteroids and ruxolitinib, which has been approved in the United States for use in acute and chronic GVHD on the basis of the REACH2 and REACH3 trials [129,130].

### 5.3. Solid Cancers and Immune Checkpoint Inhibitors (ICI)

HLH occurring in solid tumors is not common. A review of 2197 published cases of adult HLH only determined 32 cases (1.4%) to be due to solid cancers [17]. In this setting, HLH occurs in widely metastatic disease, often with marrow infiltration and underlying aggressive histologies, such as germ cell tumors or melanoma [131,132,133]. It is worth noting this incidence rate was reported prior to the broad use of immune checkpoint inhibitors (ICIs) for solid tumors, and since then, there have been many case reports of HLH occurring as an immune-related adverse event (irAE) in a variety of tumor types. The most well-studied has been in melanoma due to the widespread use of ICIs. In these early descriptions, HLH as an irAE appears to respond quickly to high-dose corticosteroids and withdrawal of the associated agent, similarly to other irAEs. It has been reported with every anti-CTLA4, -PD-1, and -PD-L1 therapy when used as monotherapy or in combinations [134,135,136]. In addition to melanoma, HLH has been reported in nearly all settings in which ICIs are approved, including Merkel cell, cutaneous squamous cell carcinoma, non-small cell lung cancer, bladder cancer, breast carcinoma, and experimentally in thymic carcinoma and glioblastoma multiforme [136,137,138,139,140,141,142,143,144].

The proposed pathophysiology of ICI-induced HLH is generally congruent with our understanding of HLH. Immune checkpoints maintain immunologic homeostasis by attenuating T cell responses to immunologic triggers; removal of these checkpoints with ICIs results in the absence of normal inhibitory control over T cell activation, causing HLH-like toxicities [145]. HLH as an irAE appears to be similar to MAS, in which high-dose methylprednisolone (1 g/day for 3–5 days) may be sufficient (with adjunctive tocilizumab or anakinra depending on the clinical context), rarely requiring etoposide unless there is an insufficient response after 48 h [24]. As anti-IL6 therapies become more commonly used for CRS, tocilizumab combined with a steroid taper may be the most effective treatment for ICI-related HLH [146]. Recent summaries on hematologic toxicities of ICIs have not shown a clear correlation between the initiation of ICI and the onset of HLH (which has been reported to occur less than one week and greater than one year after ICI initiation) [147,148,149]. A query into the World Health Organization pharmacovigilance database retrieved 38 cases of HLH and demonstrated low co-occurrence with other irAEs and infections, a median onset of 6.7 weeks after ICI initiation, with 7 cases in which HLH is the primary or contributing cause of death [150]. Notably, the recently approved LAG-3 antibody, relatlimab, was associated with one HLH death when combined with nivolumab [151]. Re-challenge with ICIs has generally not been pursued, but a small number of studies report ICI re-challenge without recurrent HLH [147,152].

### 5.4. Bispecific T Cell Engagers, CAR T Therapy, and Cytokine Release Syndrome (CRS)

Treatment-emergent cytokine storm has been repeatedly described with novel immunotherapies. These novel therapies include the bispecific antibodies (bsAbs), namely bispecific T cell engagers, which redirect CD3^+^ T cells to a target cancer antigen, and CAR T cells, which utilize autologous T cells engineered with a chimeric T cell receptor (TCR) against a target antigen. FDA/EMA-approved bsAbs with CD3 activity include blinatumomab, targeting CD19 in relapsed/refractory acute lymphoblastic leukemia (ALL); the BCMA-directed T cell engager teclistamab-cqyv for relapsed multiple myeloma; and the newly approved anti-CD20 bsAb mosunetuzumab-axgb in relapsed/refractory follicular lymphoma. Many bsAbs are in development and expected to be approved in the near future. Glofitamab and epcoritamab, both CD20/CD3 bsAbs, were shown to be effective for heavily pretreated large B cell lymphomas [153,154], and talquetamab, directed against GPRC5D in myeloma, has demonstrated efficacy in relapsed multiple myeloma [155]. There are also six CAR T therapies currently approved for clinical use, including four anti-CD19 CAR T (tisagenlecleucel, axicabtagene ciloleucel, brexucabtagene autoleucel, and lisocabtagene maraleucel) therapies to treat B cell malignancies, and two anti-BCMA CAR T (idecabtagene vicleucel and ciltacabtagene autoleucel) therapies for relapsed/refractory multiple myeloma, as well as many others being tested against new targets and in other cancers.

Cytokine release syndrome (CRS) is a well-described, treatment-emergent phenomenon with both bsAbs and CAR T therapies, in which profound cytokine elevations and HLH-like symptomatology occur from T cell activation [156,157,158]. The American Society of Transplantation and Cellular Therapy (ASTCT) has established consensus grading systems for the unique adverse effects arising from immune effector cell (IEC) therapy, namely CRS and IEC-associated neurotoxicity syndrome (ICANS) [159]. It is now increasingly recognized that HLH from IEC therapy may or may not evolve in association with CRS/ICANS but is a separate hyperinflammatory insult independent from severe CRS with HLH-like features and is usually characterized as a second inflammatory wave after an initial improvement in CRS [160,161,162].

Several groups have proposed diagnostic criteria to distinguish severe CRS from this emerging entity, called immune effector cell-associated HLH-like syndrome (IEC-HS). Neepalu et al. reported CAR-related HLH occurring in ~1% of cases, distinct from CRS in its high mortality and refractoriness to standard CRS therapy, and proposed a ferritin cutoff of >10,000 ng/mL and any two of grade >3 organ toxicities of the liver, kidney, lung, or the presence of hemophagocytosis to diagnose CAR-related HLH [49]. In a phase 1 CD22 CAR T trial, Shah et al. observed higher rates of HLH-like manifestations (33%) based on modified Neepalu criteria despite comparable rates of CRS and found these HLH manifestations occurred outside the temporal context of CRS and responded to anakinra monotherapy [163]. Lichtenstein et al. described these patients in additional detail and found HLH in 36% of patients who received CD22 CAR T, all among those who previously developed CRS; notably, coagulopathy was included as an additional diagnostic criterion [161]. The median time to CRS onset was 8 days after CAR T, and the time to HLH onset was 14 days; peak ferritin was ~10-fold higher with HLH. Kennedy et al. studied the emergence of post-BCMA CAR T MAS, defined as a ferritin increase ≥100 µg/L/h within a 24-h period and a minimum fibrinogen <150 mg/dL or maximum LDH >2 times the upper limit of normal within 14 days following CAR T and found a 22% incidence rate [164]. Several publications have confirmed these findings with other CAR T products in ALL, NHL, and multiple myeloma, with an estimated median onset of around 2 weeks after CAR T infusion but with generally poorer outcomes, including up to 67% mortality [165,166,167,168]. An ASTCT working group is expected to publish consensus criteria for IEC-HS soon, which may rely on a combination of elevated or rapidly rising ferritin; worsening inflammation after resolving, resolved, or treatment-refractory CRS; transaminase elevations; hypofibrinogenemia; hemophagocytosis; and worsening cytopenias, in addition to minor criteria assessing fever, neurotoxicity, pulmonary injury, renal insufficiency, hypertriglyceridemia, splenomegaly, hyperbilirubinemia, coagulopathy, and LDH elevations. Based on limited evidence, our front-line approach to IEC-HS is to use anakinra with or without tocilizumab and/or corticosteroids as appropriate for concurrent features of CRS/ICANS.

## 6. Conclusions

Adult HLH is a heterogenous condition with a wide range of underlying environmental triggers and/or genetic predispositions and has traditionally been reliant on pediatric studies to inform its pathophysiology and treatment outcomes. Malignancy-associated HLH has been especially poorly studied, as mHLH is uncommon in children; as such, mHLH has essentially only been described in case series and small pilot trials without definitive prospective studies, and optimal diagnostic and management strategies remain unknown. Many questions remain: do the HLH-2004 diagnostic criteria apply to mHLH or treatment-related HLH? Are the same pathophysiologic mechanisms uncovered in murine models and children the same for adults? What should be the standard of care to treat mHLH in different settings? Given these unresolved questions, it is important to emphasize that published diagnostic and treatment algorithms should only be tools and should not replace clinical reasoning, as our current approaches have substantial limitations in our ability to favorably intervene in these often quickly-decompensating patients. We advise a low threshold to screen patients suspected of having HLH with inflammatory labs as described, using a multi-step approach for the work-up of HLH, as well as adjunctive diagnostic techniques such as the HScore and OHI index.

In the end, the most prudent approach may be the simplest until stronger evidence is developed to guide clinical decision-making: does this patient have a hyperinflammatory syndrome, and if so, would this patient benefit from steroids and/or etoposide, with or without other adjunctive therapies such as ruxolitinib? Despite the unclear role of HLH-94 in mHLH, initial, aggressive strategies to dampen cytokine storm with steroids and etoposide (with ruxolitinib possibly replacing or reducing etoposide exposure) may serve to bridge patients to chemotherapy. Treatment-related HLH, such as after HCT, ICI, or IEC, does not appear to require etoposide and may be treated with corticosteroids and anti-cytokine therapies only; however, consensus diagnostic criteria will need to be established in each of these settings to distinguish between engraftment syndrome, haplostorm, GVHD, sepsis, irAEs, and CRS/ICANS, especially as cellular and immunotherapies become standard of care. Several targeted inhibitors and cellular therapies are under investigation for the treatment of HLH and associated pathologies. As data with these and other emerging approaches develop, it will be important to evaluate their integration into the management of mHLH.

## Figures and Tables

**Figure 1 cancers-15-01839-f001:**
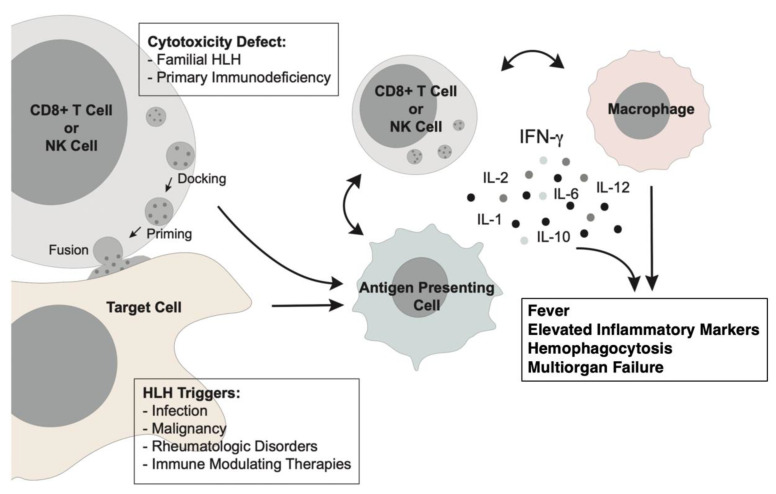
Pathophysiology of HLH [16]. Primary HLH is driven by genetic defects in T or NK cell cytotoxicity, while sHLH is driven by immune hyperactivation against an antigenic trigger. Clinical manifestations arise from a common pathway, resulting in persistent accumulation and activation of CD8^+^ T cells, NK cells, macrophages, and proinflammatory cytokines, resulting in end-organ damage. Adapted with modifications from Paolino et al. [16], with use, distribution, and reproduction permitted under the terms of the Creative Commons Attribution License (CC BY).

**Figure 2 cancers-15-01839-f002:**
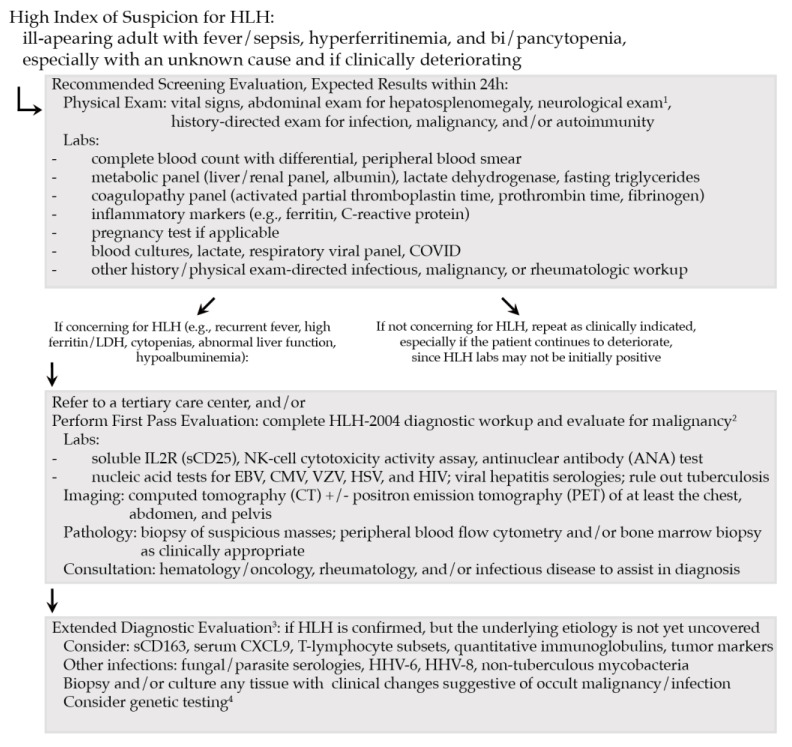
Schematic for Recommended Evaluation of Adult HLH. ^1^ Though uncommon in adults, central nervous system (CNS) manifestations of HLH are possible and should be evaluated with cerebrospinal fluid (CSF) analysis, as well as magnetic resonance imaging (MRI). ^2^ Given likelihood of malignancy in adults with HLH and its poor prognosis, mHLH must be evaluated promptly. If mHLH is diagnosed, complete cancer workup remains a priority. Biopsy confirmation of malignancy may not be possible in patients critically ill due to HLH; in these cases, we recommend proceeding with HLH-directed therapy, followed by pathologic confirmation when clinically stable. ^3^ Consider workup of other endemic causes/mimics of HLH (e.g., visceral leishmaniasis, *Rickettsia*), where appropriate, based on exposures. ^4^ Genetic testing is recommended for suspected primary HLH (young patients or family history) or patients with HLH recurrence, as HLH variants are increasingly recognized with late HLH phenotype emergence [13]. Note: treatment-associated HLH (such as with CAR-T cells or HCT) is considered a separate diagnostic entity and does not require an extensive workup as above due to the temporal proximity of HLH clinical manifestations to the cell therapy administration [49], for which the underlying cause is more obvious.

**Figure 3 cancers-15-01839-f003:**
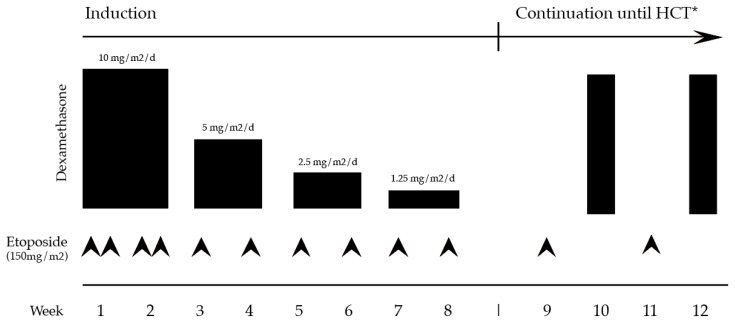
Treatment schematic of the HLH-94 protocol. * After an 8-week induction, continuation therapy is recommended for all pHLH and persistent or reactivated non-pHLH, which consists of alternating etoposide 150 mg/m^2^ every other week with dexamethasone 10 mg/m^2^/d for 3 days until HCT. Daily oral cyclosporine A was given in the original HLH-94 protocol during continuation, targeting trough levels of 200 µg/L, though this is often omitted in contemporary regimens. Reproduced with modifications from Henter et al. [53].

**Table 1 cancers-15-01839-t001:** Representative Etiologies of Adult sHLH [17].

Category	Specific Causes
Infection	
Viral	Human Herpesviridae (e.g., EBV, HSV, CMV, VZV), HIV, viral hepatitis, influenza, parvovirus B19, dengue
Bacterial	*Mycobacterium tuberculosis, Staphylococcus aureus, Rickettsia, Mycoplasma*
Parasitic	*Leishmania, Plasmodium, Toxoplasma*
Fungal	*Histoplasma, Candida, Cryptococcus, Aspergillus*
Malignancy	
Hematologic	T/NK-cell lymphomas, aggressive B-cell lymphomas, leukemia, Hodgkin lymphoma, Castleman disease
Solid	Metastatic carcinomas, sarcomas
Autoimmune	Systemic lupus erythematosus, adult-onset Still’s disease, rheumatoid arthritis, vasculitis, inflammatory bowel disease
Treatment-Related	
Transplantation	Allogeneic hematopoietic cell transplantation, solid organ transplantation
T Cell Therapy	CAR T cell therapy, bispecific T cell engagers
Other Therapy	Immune checkpoint inhibitors, chemotherapy-induced, drug-induced hypersensitivity, surgery, vaccination, hemodialysis
Other	Pregnancy, trauma, idiopathic, unknown, multifactorial

**Table 2 cancers-15-01839-t002:** HLH-2004 Diagnostic Guidelines for Hemophagocytic Lymphohistiocytosis [28].

Either:
- A molecular diagnosis consistent with HLH, or
- Five of the following eight criteria:
Fever *
2. Splenomegaly
3. Cytopenias affecting at least two lineages: neutrophils < 1.0 × 10^9^/L; hemoglobin < 10 g/dL *; platelets < 100 × 10^9^/L
4. Fasting triglycerides ≥ 3.0 mmol/L (265 mg/dL) or fibrinogen ≤ 1.5 g/L (150 mg/dL)
5. Hemophagocytosis in bone marrow, spleen, or lymph nodes *
6. Low or absent NK cell activity
7. Ferritin ≥ 500 ng/mL (reported by some labs in μg/L)
8. Soluble IL2 receptor (sIL2R, aka sCD25) ≥ 2400 U/mL *

Supporting evidence: liver biopsy showing chronic/persistent hepatitis; spinal fluid pleocytosis (mononuclear cells) and/or elevated spinal fluid protein; cerebromeningeal symptoms; liver enzyme abnormalities; lymphadenopathy; hypoproteinemia; hyponatremia; high VLDL; low HDL; skin rash; edema. * Subsequent proposed modifications/specifications to the diagnostic guidelines include fever ≥ 38.5 °C; hemoglobin < 9 g/dL except in infants; hemophagocytosis in the marrow, spleen, lymph nodes, or liver; and elevated sIL2R > 2 standard deviations from the laboratory age-adjusted mean [31].

**Table 3 cancers-15-01839-t003:** Publications on Ruxolitinib for the Treatment of Adults with mHLH.

Reference	N	Mean Age (Range)	HLH Type	New or R/R HLH	Target Rux Dose (BID)	Rux Duration	HLH Therapy	Response	OS * (f/u if Known)
Boonstra 2021 [84]	1	70	R/R Hodgkin; EBV viremia	New	15 mg	1.5 m	Rux	PR	100%
Hansen 2021 [85]	1	33	SPTL	R/R	15 mg	11 m	Dex/Etop −> Cy, Doxo, Vin, Pred −>Rux, Etop, IVIG +Alem	CR	100% (1 y)
Stalder 2023 [86]	6	52 y (34–72 y)	AML	New	10 mg	31–122 d	Dex, Etop, Rux, induction chemo	CR (83%), PR (17%)	33% (120 d)
Trantham 2020 [87]	2	66 y, 24 y	Suspected Hodgkin DLBCL	R/R	10 mg 15 mg	~6 m ~25 d	Dex/Etop −> Benda/Brentux −> Rux −> Alem/Anakinra R-EPOCH x3 −> R-CHOP x3 −> Rux, HD MTX, AraC, IT −> R-GCD −> R-ICE −> Alem/Dex	CR (100%)	0% (1 y, 14.5 m)
J Wang 2021 [88]	3	27 y, 28 y, 66 y	B cell lymphoma	R/R	10 mg	NR	HLH94 −> Rux, Doxo (lipo), Etop, Methylpred −> chemo −> HCT	NR	NR
H Wang 2020 [89]	2	24 y, 45 y	EBV+ NK cell leukemia Relapsed PTL	New	5 mg	~5 w	Dex/Etop, PLEX, Rux, Gem/Ox/Peg −> Pred Dex/Etop/Rux −> Gem/Ox/Peg	?CR (100%)	0% (~2 m?)
Zhou 2020 [90]	36	44.7 y (31–58 y)	Lymphoma	New	0.3 mg/kg daily	14 d	Dex/Etop/Rux/Doxo −> chemo	CR (28%), PR (56%)	39% (5 m)

Abbreviations: Alem, alemtuzumab; AraC, cytarabine; Benda, bendamustine; BID, twice daily; Brentux, brentuximab; CR, complete response; Cy, cyclophosphamide; Dex, dexamethasone; DLBCL, diffuse large B-cell lymphoma; Doxo, doxorubicin; EBV, Epstein–Barr virus; Etop, etoposide; Gem, gemcitabine; HD MTX, high-dose methotrexate; IT, intrathecal chemotherapy; IVIG, intravenous immunoglobulin; Lipo, liposomal; Methylpred, methylprednisolone; NR, not reported; OS, overall survival; Ox, oxaliplatin; Peg, pegaspargase; PLEX, plasmapheresis; PR, partial response; Pred, prednisone; PTL, peripheral T cell lymphoma; Rux, ruxolitinib; SPTL, subcutaneous panniculitis-like T cell lymphoma; Vin, vincristine. R/R, relapsed/refractory; f/u, follow-up; mg, milligram; kg, kilogram; d, day; w, week; m, month; y, year. * Overall survival defined as the percentage of patients alive at the time of publication with the time from initiation of ruxolitinib to the end point (last known median follow-up or death) in parentheses when available. ? Indicates uncertainty from the referenced publication.

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
