# Peer review of "Diagnosis and Management of Adult Malignancy-Associated Hemophagocytic Lymphohistiocytosis"

_cancers, 2023, doi:10.3390/cancers15061839_

Round 1
Reviewer 1 Report
The present work provides an extensive review on the diagnostic workup and treatment of malignancy-associated hemophagocytic lymphohistiocytosis (mHLH). As mHLH is a life-threatening condition associated with rapidly deteriorated clinical process and high mortality. Therefore, early diagnosis and intervention are of vital importance. In general, I think this work has some merits, but this paper is not well organized and have certain limitations:
1. The scope of this paper is too large, but the depth is not enough. The paper should give literature review on selected directions, since the goal is to emphasis on the appropriate workup and treatment of mHLH. Since mHLH can be divided into malignancy-induced HLH, infection-induced HLH and immunotherapy-related HLH, treatment strategy varies with different subtypes. Please further elaborate on possible treatment options for different subtypes to avoid misunderstanding. For example, the role of anti-IL6 agents in malignancy-induced HLH is still unclear.
2. The diagnostic workup is not well illustrated, and a schema of diagnostic approach is suggested.
3. Some of the content is more straightforward to illustrate with tables and graphs, such as pathogenesis of mHLH, aetiology of HLH, etc.
Author Response
Reviewer 1:
The present work provides an extensive review on the diagnostic workup and treatment of malignancy-associated hemophagocytic lymphohistiocytosis (mHLH). As mHLH is a life-threatening condition associated with rapidly deteriorated clinical process and high mortality. Therefore, early diagnosis and intervention are of vital importance. In general, I think this work has some merits, but this paper is not well organized and have certain limitations:
1. The scope of this paper is too large, but the depth is not enough. The paper should give literature review on selected directions, since the goal is to emphasis on the appropriate workup and treatment of mHLH.
We thank the reviewer for his/her comments and agree that the paper would benefit from concision. As such, we have removed/consolidated several paragraphs of the paper to focus specifically on diagnosis and treatment (e.g., the introduction in Section 1 has been reduced from 6 paragraphs and 1094 words, to 4 paragraphs and 522 words; Section 3 has been predominantly consolidated into Figure 2). The text word count has decreased from 7744 to 6321. Selected directions of the review have been emphasized in Section 5, which describe, in detail, the current literature on four distinct etiologies of mHLH most likely encountered by the adult oncologist.
Since mHLH can be divided into malignancy-induced HLH, infection-induced HLH and immunotherapy-related HLH, treatment strategy varies with different subtypes. Please further elaborate on possible treatment options for different subtypes to avoid misunderstanding.
We agree that malignancy-associated HLH (mHLH) can be subdivided in various ways, though none of these subdivisions are well-established in the literature or by consensus. We do not consider infection-induced HLH a subdivision of mHLH, as it is not related to cancer unless in setting of virally-driven lymphomas, which is discussed in the lymphoma section (Section 5.1), or unless related to immunosuppression from cancer-directed therapies, which is not particularly distinct from infection-induced HLH in general. Instead, in Section 5, we have chosen special categories of mHLH to discuss that are most interesting to the reader – lymphoma, leukemia/hematopoietic cell transplantation, solid cancers/immunotherapy, and T cell therapies receive particular attention from the practicing oncologist. Additionally, the approaches to HLH in the setting of allogeneic transplant, checkpoint inhibition, and T cell therapies too different to be grouped altogether within “immunotherapy-related HLH,” and thus discussed separately (Sections 5.2, 5.3, and 5.4).
For example, the role of anti-IL6 agents in malignancy-induced HLH is still unclear.
We agree that anti-IL6 agents for mHLH is unclear, as are essentially all agents for mHLH due to the lack of a standard of care. However, tocilizumab has been evaluated in a few publications (Posas-Mendosa et al., Rheumatology (Oxford) 2021; Noveihed et al., Discov Oncol 2022; Kim et al., Orphanet J Rare Dis 2022), recognizing these are case reports. However, given its widespread use in MAS, CRS, COVID, and other inflammatory disorders such as sJIA, we anticipate readers to be interested in a summary of the current literature regarding anti-IL6 for mHLH.
2. The diagnostic workup is not well illustrated, and a schema of diagnostic approach is suggested.
We have added a schematic to illustrate the diagnostic workup, under Section 3 (“Figure 2”).
3. Some of the content is more straightforward to illustrate with tables and graphs, such as pathogenesis of mHLH, aetiology of HLH, etc.
We added a figure to illustrate the pathophysiology of HLH (“Figure 1”) and a table with the most common etiologies of HLH (“Table 1”).
Reviewer 2 Report
This good review review addresses quite extensively the subject, citing adequately the available literature and, when appropriate, pointing to unclear or unsettled issues, which are not few. Due to the complexity of the diagnosis and treatment of HLH and, particularly, of mHLH and since the review is clearly practice-oriented, it would have been helpful to see in it, a graphical outline of the authors' suggestion of workup and conduct. Though not necessary, such a graphical abstract would be very helpful to the practitioner facing patients with suspected mHLH.
Author Response
Reviewer 2:
This good review review addresses quite extensively the subject, citing adequately the available literature and, when appropriate, pointing to unclear or unsettled issues, which are not few. Due to the complexity of the diagnosis and treatment of HLH and, particularly, of mHLH and since the review is clearly practice-oriented, it would have been helpful to see in it, a graphical outline of the authors' suggestion of workup and conduct. Though not necessary, such a graphical abstract would be very helpful to the practitioner facing patients with suspected mHLH.
We appreciate the reviewer’s suggestions and appreciation of the complexity and unsettled questions around HLH. To substitute for a graphical abstract (given the complexity of the topic), we have provided several new figures/tables to better illustrate the content of the paper for the reader, including a figure illustrating the pathophysiology of HLH (“Figure 1”), a table with the most common etiologies of HLH (“Table 1”), and a schematic to illustrate our approach to HLH workup (“Figure 2”).
Reviewer 3 Report
Lee JC and Logan AC have done an extensive review of adult malignancy-associated HLH. This review is well done but too long and complicated for the reader. It lacks simplicity to understand the medical challenge in the diagnosis and management of HLH associated with malignancy in adults.
1. In the introduction, the pathophysiology described can be confusing. It needs to be shortened and more precise.
A. Primary HLH (more common in children) are genetic diseases linked to a deficit in lymphocyte cytotoxicity. In this context, there is a persistence of the trigger (essentially viral), resulting in a lymphoproliferative disease and then, a cytokine storm (hyperinflammation). Tumoral syndrome (adenopathy, splenomegaly, hepatomegaly) and neurological symptoms are secondary to lymphoproliferation.
fHLH (lineage 51) is a subtype of primary (genetic) HLH.
B. Secondary HLH (more common in adults) has been associated with hypomorphic mutations in lymphocyte cytotoxicity but the results are controversial and secondary HLH may be more a direct consequence of a cytokine storm than lymphoproliferation (Les functional and genetic tests in adults with HLH reveal cytotoxicity defect rather than Blood 136, 542–552 (2020)) apart from lymphoid malignancies.
Lines 96-97: allogeneic HSCT is not a current indication of secondary HLH. In any case the reader should not consider it as an easy possibility.
Line 100-105: the list of adverse events of corticosteroid therapy is not relevant here
C. Moreover, the diagnosis and the therapeutic issues must be specified at the end of the introduction.
The diagnosis of malignancy is an absolute emergency if the HLH is inaugural, especially for hematological cancers such as lymphomas.
If the malignancy is already known and treated, HLH is more often the consequence of antitumor therapies. Adverse immune events but also secondary infectious complications.
An exhaustive search for opportunistic microorganisms (bacteria, mycobacteria, exhaustive screening for viruses including all viruses of the herpes group, fungi, parasites) must be done before considering immunosuppressive therapies. These patients are severely immunocompromised.
Line 42-43: “Inflammasome homeostasis is impaired by an inability to terminate the immune response [12]. This sentence is not based on solid scientific evidence.
2. Diagnosis of malignancy-associated HLH
A figure is recommended here to clarify what reasoning the physician should have.
The elements useful for the diagnosis of HLH must be distinguished from the elements useful for the diagnosis of malignancy in two distinct paragraphs and thus, parts 2 and 3 (initial assessment of undifferentiated adult HLH) could be merged.
Among the biological abnormalities suggestive of HLH, in addition to cytopenias, ferritin, acute hepatitis is also a possibility.
There are also two main points missing:
- The malignancy is known and treated:
In an immunocompromised patient, there is an absolute need to eliminate an opportunistic infectious complication: exhaustive viral screening by PCR in blood and tissues, blood cultures, research for mycobacteria, serology, etc.
If the patient is receiving immunotherapy, an immune adverse event should be considered.
- When the malignancy is not yet known, an aggressive diagnostic strategy is very important to start specific antitumor treatment as soon as possible.
The HLH-2004 criteria are not very interesting here.
Authors should explain the value of sCD25 assay and monitoring: T cell proliferation
The authors must explain the interest of the dosage and monitoring of ferritin: activation of macrophages
3. Treatment of mHLH
In the case of an inaugural mHLH, the specific treatment of the malignancy is an absolute emergency. Again, there is a big difference if the patient is already treated and immunocompromised or if immunotherapy (CPI, CAR T cells) is in progress !
The authors must specify which specific therapy could be envisaged in particular conditions: lymphoma, leukaemia, solid cancers. Even if only case reports or case series exist, the most relevant treatments should be retained. Therapies such as emapalumab, ELA026 are not relevant here. By this I mean that the manuscript should be shortened and simplified.
Line 664: To the best of my knowledge, T cell hyperproliferation has never been documented in CRS ?
Author Response
Reviewer 3:
Lee JC and Logan AC have done an extensive review of adult malignancy-associated HLH. This review is well done but too long and complicated for the reader. It lacks simplicity to understand the medical challenge in the diagnosis and management of HLH associated with malignancy in adults.
We thank the reviewer for his/her comments and agree that the review benefits from being somewhat shorter. As such, we have removed/consolidated several paragraphs of the paper to focus specifically on diagnosis and treatment (e.g., the introduction in Section 1 has been reduced from 6 paragraphs and 1094 words, to 4 paragraphs and 522 words; Section 3 has been predominantly consolidated into Figure 2). To improve clarity, we have added several subheadings particularly to Section 4 to serve as guideposts for the reader. The text word count has decreased from 7744 to 6321.
1. In the introduction, the pathophysiology described can be confusing. It needs to be shortened and more precise.
For clarity and concision, we have greatly reduced the word count and added a figure to illustrate the pathophysiology for HLH. We agree that this section should not be emphasized since describing HLH pathophysiology is not a major aim of this review. Pathophysiology is now consolidated into one figure and one paragraph, as follows:
“Current understanding of HLH pathogenesis is derived from murine models and primary patient samples. In these studies, CD8+ T cells have been shown to be activated in response to an immunologic trigger, leading to production of type 2 interferon (IFN) that primes macrophages to secrete additional proinflammatory cytokines (Figure 1) [3–7]. Deficiencies in this cytolytic pathway result in an inability to proceed with normal activation-induced cell death, generating uncontrolled accumulation and activation of CD8+ T cells, natural killer (NK) cells, macrophages, and proinflammatory cytokines [8]. When HLH occurs as the result of congenital deficiencies of key cytolytic pathway proteins, this is called primary HLH (pHLH), which mainly occur in children [9,10]. In adults, HLH is usually driven by a highly immunogenic trigger (secondary HLH, sHLH) rather than primary cytotoxicity defects [11], though in some cases, a relevant, often hypomorphic genetic mutation affecting cell-mediated immunity may be identified [12–15].”
A. Primary HLH (more common in children) are genetic diseases linked to a deficit in lymphocyte cytotoxicity. In this context, there is a persistence of the trigger (essentially viral), resulting in a lymphoproliferative disease and then, a cytokine storm (hyperinflammation). Tumoral syndrome (adenopathy, splenomegaly, hepatomegaly) and neurological symptoms are secondary to lymphoproliferation.
fHLH (lineage 51) is a subtype of primary (genetic) HLH.
We de-emphasized familial and primary HLH, and all references to a genetic form of HLH have been consolidated as primary HLH (pHLH). Clinically evident lymphoproliferation, though common, is not a requirement for HLH, and of the humoral syndromes mentioned, only splenomegaly is considered in the HLH-2004 diagnostic criteria (Henter et al., Pediatr Blood Cancer 2007); thus, we have not emphasized a resulting lymphoproliferative disorder in the context of HLH.
B. Secondary HLH (more common in adults) has been associated with hypomorphic mutations in lymphocyte cytotoxicity but the results are controversial and secondary HLH may be more a direct consequence of a cytokine storm than lymphoproliferation (Les functional and genetic tests in adults with HLH reveal cytotoxicity defect rather than Blood 136, 542–552 (2020)) apart from lymphoid malignancies.
We agree that secondary HLH most often develops from strong immunological activation rather than obvious cytotoxic defects. We thank the reviewer for the additional insight provided and have added the publication cited by the reviewer. The following sentence has been clarified:
“In adults, HLH is usually driven by a highly immunogenic trigger (secondary HLH, sHLH) rather than primary cytotoxicity defects [11], though in some cases, a relevant, often hypomorphic genetic mutation affecting cell-mediated immunity may be identified [12–15].”
Lines 96-97: allogeneic HSCT is not a current indication of secondary HLH. In any case the reader should not consider it as an easy possibility.
We agree that allogeneic hematopoietic cell transplant for secondary HLH is rare and not indicated except in the relapsed/refractory setting. This has been supported in publications such as Trottestam et al., Blood 2011., which established HLH-94 and allogeneic transplant in pediatric HLH, and the Histiocyte Society recommendations for adult HLH per La Rosée et al., Blood 2019. Several subsequent studies enrolling relapsed/refractory, non-genetic HLH for allogeneic HCT have been conducted, such as Allen et al., Blood 2018, and Gooptu et al., Blood Advances, 2022.
Line 100-105: the list of adverse events of corticosteroid therapy is not relevant here
This and the subsequent lines regarding HLH-94-related toxicities have been removed.
C. Moreover, the diagnosis and the therapeutic issues must be specified at the end of the introduction.
Since diagnosis and therapy are discussed at length in subsequent sections, we elect to minimize redundancy by only introducing the major questions concerning diagnosis and treatment in the introduction and elect to specify these issues in greater detail in subsequent sections.
“There has not yet been a large prospective study to validate treatment strategies for adults with newly diagnosed HLH. Based on retrospective data, outcomes in adults differ widely between non-malignant (nmHLH) and malignancy-associated (mHLH)…It is unknown, however r, whether treatment of mHLH confers a survival benefit over treatment of the underlying malignancy. It also is unknown whether the diagnostic criteria for mHLH should be the same as with nmHLH. This review provides an updated summary of the existing literature on the diagnosis and management of adult malignancy-associated HLH, especially in the setting of emerging research on effective treatment strategies in the age of engineered cellular and immunotherapies.”
The diagnosis of malignancy is an absolute emergency if the HLH is inaugural, especially for hematological cancers such as lymphomas.
We agree completely and this has been emphasized in several instances within Section 3, such as:
“Given the nontrivial turnaround time for several laboratory tests specified in HLH-2004, early suspicion and diagnosis is critical to allow for therapeutic intervention prior to organ failure and death that commonly occurs rapidly in adults with mHLH.”
“…Underlying malignancy should be thoroughly evaluated and even expected, particularly for older adults presenting with HLH, with a low threshold to pursue advanced imaging.”
“In fact, because the window between safely pursuing a tissue biopsy and clinical deterioration can be very short (sometimes within a day), earlier imaging and biopsy may be warranted.”
“3Given high likelihood of malignancy in adults with HLH and its poor prognosis, mHLH must be evaluated promptly due to the necessity of emergent cancer-directed therapy. If mHLH is diagnosed, complete cancer workup per national guidelines or standard clinical practice remains a priority. Biopsy confirmation of malignancy may not be possible, however, in patients critically ill due to HLH; in these cases, we recommend proceeding with HLH-directed therapy, followed by pathologic confirmation when clinically stable.”
If the malignancy is already known and treated, HLH is more often the consequence of antitumor therapies. Adverse immune events but also secondary infectious complications. An exhaustive search for opportunistic microorganisms (bacteria, mycobacteria, exhaustive screening for viruses including all viruses of the herpes group, fungi, parasites) must be done before considering immunosuppressive therapies. These patients are severely immunocompromised.
We agree; a comprehensive infection evaluation has been extensively described in Section 3, which discusses the workup of undifferentiated HLH. Cases in which an underlying malignancy is known (treatment-related HLH) are discussed in Section 5.2, 5.3, and 5.4. Because the presentation of these cases to the treating clinician is clearly distinct from an inaugural presentation of HLH, these cases are discussed in separate sections.
Line 42-43: “Inflammasome homeostasis is impaired by an inability to terminate the immune response [12]. This sentence is not based on solid scientific evidence.
In our effort to simplify discussion of HLH pathogenesis, we have removed this statement.
2. Diagnosis of malignancy-associated HLH
A figure is recommended here to clarify what reasoning the physician should have.
We have added a schematic to illustrate the diagnostic workup, under Section 3 (“Figure 2”).
The elements useful for the diagnosis of HLH must be distinguished from the elements useful for the diagnosis of malignancy in two distinct paragraphs and thus, parts 2 and 3 (initial assessment of undifferentiated adult HLH) could be merged.
The aim of Section 2 is to discuss the varying methods and challenges in diagnosing mHLH, especially as there are no established diagnostic algorithms. The aim of Section 3 is to provide practical guidance for the clinician in working up a new suspected HLH diagnosis in light of these diagnostic uncertainties. Because of these differing aims, we have elected to keep these sections separate.
Among the biological abnormalities suggestive of HLH, in addition to cytopenias, ferritin, acute hepatitis is also a possibility.
We agree that hepatitis is common, and this is noted in Section 2 Table 2. We elect to emphasize the Histiocyte Society revised HLH-2004 consensus diagnostic guidelines (Henter et al., Pediatr Blood Cancer 2007), which is most used in clinical practice.
There are also two main points missing:
- The malignancy is known and treated:
In an immunocompromised patient, there is an absolute need to eliminate an opportunistic infectious complication: exhaustive viral screening by PCR in blood and tissues, blood cultures, research for mycobacteria, serology, etc.
This has been elaborated at length in Section 3, including in Figure 2 with several instances of infectious evaluation, but additionally as follows:
“Note: treatment-associated HLH (such as with CAR-T cells or HCT) is considered a separate diagnostic entity and do not require an extensive workup as above due to the temporal proximity of HLH clinical manifestations to the cell therapy administration [46], for which the underlying cause is more obvious. However, given these patients are often severely immunocompromised, an exhaustive search for opportunistic infections should be performed.”
If the patient is receiving immunotherapy, an immune adverse event should be considered.
A dedicated discussion on immunotherapy is provided in Section 5.3.
When the malignancy is not yet known, an aggressive diagnostic strategy is very important to start specific antitumor treatment as soon as possible.
We agree completely and this has been emphasized in several instances within Section 3, as indicated previously.
The HLH-2004 criteria are not very interesting here.
The HLH-2004 diagnostic criteria are commonly used to diagnosis adult mHLH (e.g., Lehmberg et al., Haematologica 2015; Daver et al., Cancer 2017), but we agree that many alternative diagnostic algorithms have been proposed, such as the HScore and the OHI index. These diagnostic uncertainties are elaborated in Section 2, paragraphs 3-6, as there is ongoing debate regarding how to diagnose HLH in this setting.
Authors should explain the value of sCD25 assay and monitoring: T cell proliferation
The authors must explain the interest of the dosage and monitoring of ferritin: activation of macrophages
We have included this additional insight in Section 2 paragraph 1, citing the original papers supporting the use of ferritin and sCD25 for HLH (Esumi et al., Acta Paediatrica 1989; Komp et al., Blood 1989).
“In patients without an HLH-predisposing genetic variant, five of eight diagnostic criteria are required (Table 2), some of which denote macrophage activation such as ferritin elevation and hemophagocytosis, and some which denote T cell proliferation such as soluble IL2 receptor (sIL2R) [43,44].”
3. Treatment of mHLH
In the case of an inaugural mHLH, the specific treatment of the malignancy is an absolute emergency. Again, there is a big difference if the patient is already treated and immunocompromised or if immunotherapy (CPI, CAR T cells) is in progress !
We agree completely and this has been emphasized in several sections as indicated previously, with Sections 5.3 and 5.4 dedicated to CPI and CAR T.
The authors must specify which specific therapy could be envisaged in particular conditions: lymphoma, leukaemia, solid cancers. Even if only case reports or case series exist, the most relevant treatments should be retained. Therapies such as emapalumab, ELA026 are not relevant here. By this I mean that the manuscript should be shortened and simplified.
We thank the reviewer for her/his suggestion. The intent of Section 4 is to discuss both common and investigational agents for the treatment of HLH. As the reviewer can appreciate, there have been no clinical trials establishing standard of care for adult mHLH, so the only data available are case series, pilot studies, and retrospective cohort studies. Due to this, therapies such as emapalumab are relevant since they have been used clinically for mHLH (Johnson et al., Blood 2022). We posit that a discussion of investigational therapies such as ELA026 is warranted given the dearth of active agents. Specific agents (if available) for lymphoma, leukemia, and solid cancers are described in Section 5.
Line 664: To the best of my knowledge, T cell hyperproliferation has never been documented in CRS ?
Therapeutic T cell proliferation has been associated with CRS in both CD19 CAR T and T cell-engaging therapies such as in the review by Maude et al., Cancer J 2014. However, we agree that T cell activation is the more significant contributor to CRS, so have changed this sentence as such:
“Cytokine release syndrome (CRS) is a well-described, treatment-emergent phenomena with both bsAbs and CAR-T therapies, in which profound cytokine elevations and HLH-like symptomatology occur from T cell activation.”
Round 2
Reviewer 3 Report
Dear authors,
Congratulations on your work. It is very interesting and well documented.
I think it's still too long :) and that you must insist again on opportunistic infections-induced HLH.
This will be the only nuance.
Thank you for your confidence.